# Metamers of neural networks reveal divergence from human perceptual systems

**Jenelle Feather**[1,2,3]   **Alex Durango**[1,2,3]   **Ray Gonzalez**[1,2,3]   **Josh McDermott**[1,2,3,4]
[1] Department of Brain and Cognitive Sciences, Massachusetts Institute of Technology
[2] McGovern Institute, Massachusetts Institute of Technology
[3] Center for Brains Minds and Machines, Massachusetts Institute of Technology
[4] Speech and Hearing Bioscience and Technology, Harvard University
`{jfeather,durangoa,raygon,jhm}@mit.edu`

## Abstract

Deep neural networks have been embraced as models of sensory systems, instantiating representational transformations that appear to resemble those in the visual and auditory systems. To more thoroughly investigate their similarity to biological systems, we synthesized model metamers – stimuli that produce the same responses at some stage of a network's representation. We generated model metamers for natural stimuli by performing gradient descent on a noise signal, matching the responses of individual layers of image and audio networks to a natural image or speech signal. The resulting signals reflect the invariances instantiated in the network up to the matched layer. We then measured whether model metamers were recognizable to human observers – a necessary condition for the model representations to replicate those of humans. Although model metamers from early network layers were recognizable to humans, those from deeper layers were not. Auditory model metamers became more human-recognizable with architectural modifications that reduced aliasing from pooling operations, but those from the deepest layers remained unrecognizable. We also used the metamer test to compare model representations. Cross-model metamer recognition dropped off for deeper layers, roughly at the same point that human recognition deteriorated, indicating divergence across model representations. The results reveal discrepancies between model and human representations, but also show how metamers can help guide model refinement and elucidate model representations.

## 1 Introduction

Artificial neural networks now achieve human-level performance on tasks such as image and speech recognition, raising the question of whether they should be taken seriously as models of biological sensory systems [1, 2, 3, 4, 5]. Detailed comparisons of network performance characteristics in some cases reveal human-like error patterns, suggesting computational similarities with humans [6, 7, 8]. Other studies have found that brain responses can be better predicted by features learned by deep neural networks than by those of traditional sensory models [2, 8]. On the other hand, neural network models can typically be fooled by adversarial perturbations that have no effect on humans [9, 10], are in some cases excessively dependent on particular image features, such as texture [11], and do not fully mirror human sensitivity to image distortions [12, 13], suggesting differences with human perceptual systems. However, these discrepancies have primarily been demonstrated using stimuli specifically constructed to induce classification errors. Here, we demonstrate that the divergence between artificial network and human representations occurs generically rather than only in adversarial situations.

We use "model metamers" to test the similarity between human and artificial neural network representations. Metamers are stimuli that are physically distinct but that are perceived to be the same by an observer. Stimuli that are metameric for humans have long been used to infer the underlying structure of the human perceptual system. Metamers provided some of the original evidence for trichromacy in human color vision, and have also been applied to texture perception [14] and visual crowding [15, 16]. Related ideas can also be used to test models of neural computation [17]. Here we leverage the idea that metamers for a valid model of human perception should also be metamers for humans. Model metamers produce the same activations in a model layer as some other stimulus (here a natural sound or image). Because the activations at all subsequent layers must also be the same, the metamers are classified the same by the model. Here, we approximate model metamers via iterative optimization, producing stimuli that produce nearly the same activations as a natural stimulus, thus leading to the same network prediction. As a test of whether the model accurately reflects human perception, we measure whether humans also correctly classify the model metamers. Although this test is looser than the classical metamer test (which requires metamers to be fully indistinguishable), it is conservative with respect to the goal of testing a model of human recognition. We consider model metamers that are unrecognizable to a human to be a model failure, cognizant that models that do not perfectly match human representations in this way might nonetheless be useful in other respects.

Because the neural network models we consider are trained to classify exemplars of highly variable object or speech classes, and thus to instantiate representations that are invariant to within-class variation, it is expected that metamers from deeper layers will exhibit greater physical variability than those from early layers. The question we sought to answer is whether the nature of the invariances would be similar to those of humans, in which case the model metamers should remain human-recognizable regardless of the stage from which they are generated. We generated model metamers for three image-trained and five sound-trained models that perform well on state-of-the-art tasks and then measured human recognition of the model metamers in psychophysical experiments. We also applied the same method across networks, to ask whether the invariances learned by one network resemble those learned by another. The results establish metamers as a tool to test and understand deep neural networks, with potential uses for multi-task applications, transfer learning, and network interpretability.

## 2 Related Work

### 2.1 Visualization of deep networks

Previous neural network visualizations have used gradient descent on the input signals to visualize the representations in neural networks [18], in some cases matching the activations at a given layer [19] as we do here. Natural image priors have been shown to make images reconstructed in this way "look" more natural, and further regularization tools have been proposed with a similar purpose [20, 21]. Although such regularization can generate visually appealing images, the importance of using a natural image prior suggests differences between the network representations and those of humans. Taking this observation as a starting point, we measured the human-recognizability of images or sounds that were matched at different network stages without imposing a separate prior, to quantify the potential divergence in representations and get clues as to its origins.

### 2.2 Comparing networks with other networks

Prior work on network similarity relates the learned representations via methods such as canonical correlation analysis (CCA) [22, 23, 24]. Other such work has been inspired by the neuroscience technique of representational similarity analysis [25, 26]. Here we also use metamers for model comparison, on the grounds that metamers for one model should also be metamers for another model (as measured here by producing the same class labels, although one could apply more fine-grained methods) if the two models share invariances.

### 2.3 Metamers applied to averaged features

Metamers have been used to develop models of human perception by pooling features to directly induce invariance across space or time. Work on visual crowding used images that have the

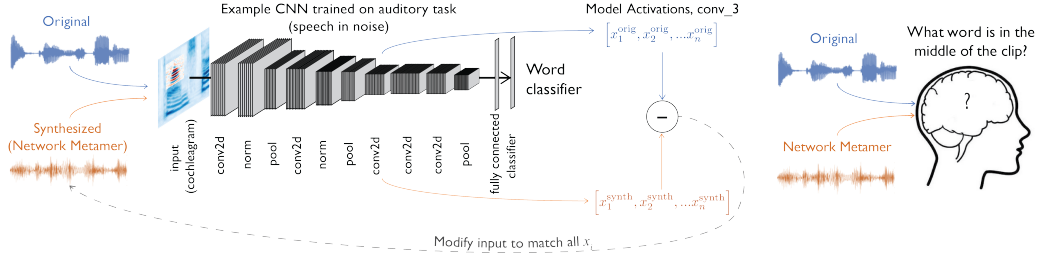

Figure 1: Model metamers are constructed by optimizing a random input signal such that it matches the measured activations of an original signal at a particular network stage. Model metamers are then presented to humans (or other networks) to measure the similarity of internal representations.

same spatially-averaged statistics in the periphery and are indistinguishable from the original in particular viewing conditions [27, 16, 28, 29]. Other work has used time-averaged statistics measured from auditory models, generating auditory textures that are mistaken for the original natural sound [30, 31]. Our work here is a more general instantiation of the metamerism approach, applicable to domains outside of peripheral vision and texture where invariances arise in the service of recognition rather than as a direct consequence of pooling.

## 3    Methods

### 3.1    Metamer generation

Model metamers were generated using an iterative feature visualization technique [19] [1]. We initialized the metamer with noise and then performed gradient descent to minimize the squared error between its network activations and those for a paired natural signal. All models and metamer generation were implemented in TensorFlow [32]. Metamer synthesis used 15000 iterations of the Adam optimizer [33] with a learning rate of 0.001, with the exception of the VGGish Embedding (0.01) and DeepSpeech (0.0001) models.

In order to validate that we had appropriately matched the synthetic signal to the original, we computed the Spearman correlation between the model metamer and corresponding original signal. These correlations were typically close to 1 (Figure 2). Once candidate metamers were generated, the following two conditions had to be true for a model metamer to be included in our experiments: (1) The network predicted the same label for the synthetic metamer and the paired natural image. This is the same classification test we apply to humans and other networks. (2) The Spearman $\rho$ between the metamer and natural image fell outside of a null distribution measured between 1,000,000 randomly chosen image or audio pairs from the training set. We compare to a null distribution rather than applying a strict threshold because the expected correlation varies with the network and layer. Setting hard cutoffs could potentially call samples metameric which are no more matched than chance, and we empirically found this procedure crucial for the random network (Figure S3). Histograms of the null and metamer correlations for all networks and selected layers are included in Tables S4-S5 and Figures S1-S8.

We found empirically that it was difficult to match some layers after a ReLU activation due to the initialized signal producing many activations of zero (Fig 2(b)). To improve the optimization, we modified the gradient through the metamer generation layer ReLU to be 1 for all values, including for values below zero, when generating a metamer for activations immediately following a ReLU. Figure 2(c) shows the matching fidelity (as measured by Spearman's $\rho$) for 20 example metamers generated with either the normal gradient or the modified gradient. The modified gradient substantially improved the matching on some layers (layer_3 of DeepSpeech, and conv_4 of the Word Trained CNN). We used the modified gradient for all metamers generated after a ReLU.

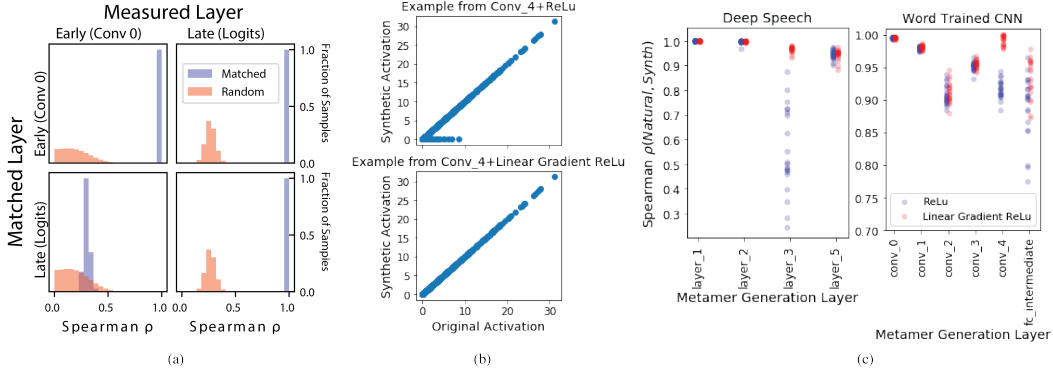

Figure 2: Validation of model metamer optimization (a) The model metamer is intended to produce the same activations as the original stimulus in a particular network layer. We quantified the fidelity of the matching as the Spearman correlation between the activations produced by a model metamer and the corresponding original stimulus, with histograms across stimuli. For a comparison null distribution, we also measured the correlation for randomly chosen pairs of signals from the training set. As intended, metamers generated from an early layer (top row) are well matched to the original in the early layer, with correlations close to 1 (blue distribution, top left), far above the null distribution across stimuli (red). Because the networks used here are deterministic and feedforward, the metamers should also produce the same activations at all subsequent layers, and they do (correlations near 1 in late layers, blue distribution, top right). Because of the many-to-one mapping instantiated by the network, metamers for a late layer (bottom row) do not match the activations in the early layer better than chance (left), but match the late layer as intended (right). (b) Comparison of activation matching with a standard ReLU activation function gradient and with the modified ReLU gradient. Without the modification, many non-zero values in the original activation get matched to zero. (c) Example layer-wise matching fidelity for metamers generated with either the standard ReLU gradient (blue) and the linear gradient ReLU (red) for two audio networks. In both networks there are layers that are significantly better matched using the modified ReLU gradients.

For visual metamers, pixel values were bounded between 0-255 or 0-1 (matching the preprocessing of the trained network), and were initialized with white noise with mean at the center value of the range. No other regularization was employed. For audio metamers, we applied gradient clipping to operations that resulted in problems with the optimization (specifically, logarithms and power operations) which were present in the audio pre-processing (that transformed the waveform to a frequency representation that provided the input to the networks). The audio metamer generation was initialized with pink noise at an RMS value of 0.01.

## 3.2 Auditory models

Our experiments used a five-layer convolutional network trained on the output of a model of the human ear. This cochlear model consisted of a filterbank of 171 filters spaced between 20Hz-80Hz with bandwidths and spacing modeled on the human ear [34, 30]. The envelope of each resulting audio subband was extracted via the Hilbert transform, downsampled to 200Hz, and passed through a compressive non-linearity. This yielded a 'cochleagram' representation, similar to a conventional spectrogram but with frequency resolution based on the human cochlea. We trained an architecture similar to that in [8] (full architecture described in Table S2).

Many neural networks do not obey the sampling theorem (because downsampling occurs without a preceding lowpass filter), and others have suggested that this could yield invariances that do not align with human perception [35, 36, 37]. Motivated by these observations, we constructed a modified architecture to reduce aliasing artifacts (Table S3). The modifications replaced max pooling operations with weighted average pooling using a hanning kernel applied with stride equal to that of the original max pooling. Any convolutional layer with a stride greater than one was replaced with a convolutional layer with a stride of one, followed by a hanning pooling operation with stride equal to the original convolutional stride.

As a demonstration that model metamers could be used to investigate representations in other audio models, we also generated example metamers from the VGGish network, which outputs embeddings used for training an environmental sound classifier and was released with the AudioSet dataset [38]. We also generated metamers for the publicly available DeepSpeech architecture [39].

### 3.3 Auditory CNN training

The auditory models were trained a word recognition task similar to [8], using segments from the Wall Street Journal [40] and Spoken Wikipedia Corpora [41]. Two-second speech segments were used for training examples, with the word in the middle of the clip assigned as the class label for training. There were 793 word classes sourced from 432 unique speakers, with 230357 unique clips in the training set and 40651 segments in the validation set (full details of the dataset construction are in Section S1.1). During training, the speech segments were randomly shifted in time and superimposed on a subset of 718625 AudioSet examples, spanning 516 AudioSet categories [42]. Some CNN models were trained to predict the AudioSet labels. In order to match performance between multiple models trained on the same task in Section 4.3 and eliminate confounds due to task performance, we used an early stopping criteria on the validation set of 57% correct for the word task and a mean area under the curve (AUC) of 0.83 for the AudioSet task.

### 3.4 Auditory metamer generation and experiments

We measured human recognition of model metamers using a task similar to that of [8]. A human observer listened to a clip and chose one of 587 possible word labels. Sixteen participants completed the experiment, each completing five trials from each of the included conditions, randomly ordered. Stimuli were generated from a set of 295 speech exemplars from the WSJ corpus (see Table S1.2 for a summary of auditory model metamers, and Figures S1-S5 for full histograms of the null and metamer Spearman $\rho$). Five sets of CNN metamers were generated for the experiment, one for each of five models: 1) the architecture inspired by [8], trained on the word task, 2) the random initialization of the reduced aliasing architecture, and 3) the reduced aliasing architecture trained on the word task, 4) the reduced aliasing architecture trained on the AudioSet task, and 5) the reduced aliasing architecture trained simultaneously on the AudioSet and word tasks. For each model, we included metamers constructed by matching the representations of the activation following each convolutional layer, fully-connected layer, and the logits (with the exception of the hanning pooling layer in the reduced aliasing networks that immediately followed strided convolutions, to equate the number of features to that for the aliasing networks). We also included metamers for the cochlear representation.

### 3.5 Image models, metamer generation, and experiments

ImageNet-trained models were obtained from publicly available pretrained checkpoints[2]. We generated metamers from a subset of layers for each of VGG-19 [43], Inception-V3 [44], and ResNet-101-V2 [45]. To compare performance between networks and humans in the visual domain, we used a modified version of the image classification task described in [13]. For each of a set of layers in the three pretrained ImageNet models, we generated metamers of 36 randomly selected natural images across each of the 16 MS-COCO categories (see Supplement Table 4 for a summary of matching the visual model metamers, and Supplement Figures 1-3 for full histograms of the null and metamer histograms). Each of sixteen participants had to classify a subset of these metameric stimuli and their corresponding natural image seeds, choosing the MS-COCO category; each participant classified 10 examples per network-layer metamer condition.

## 4 Results

### 4.1 Image network model metamers

For all tested image networks,the metamers became unrecognizable to humans by the final stages of the network (Figure 3a-b). The appearance of the metamers to humans varied depending on the architecture. In Inception-V3 and ResNet-101-V2 (both of which include convolutions with

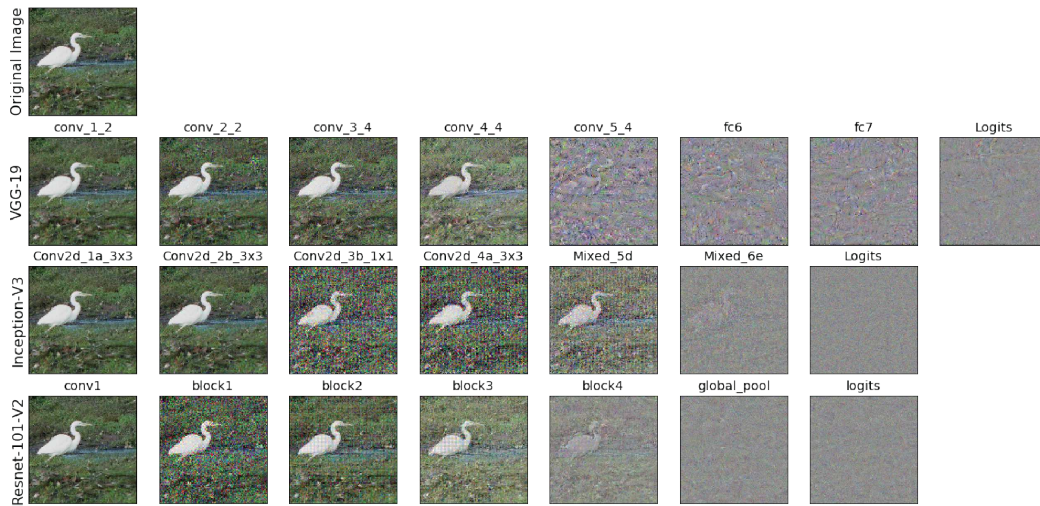

(a)

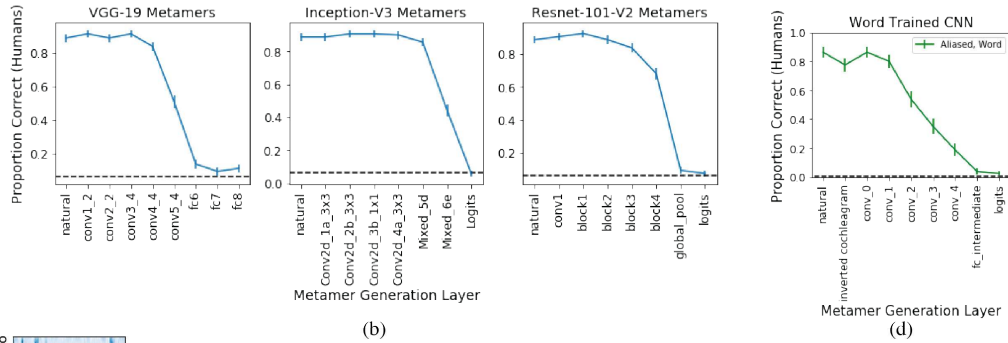

(b)                                                    (d)

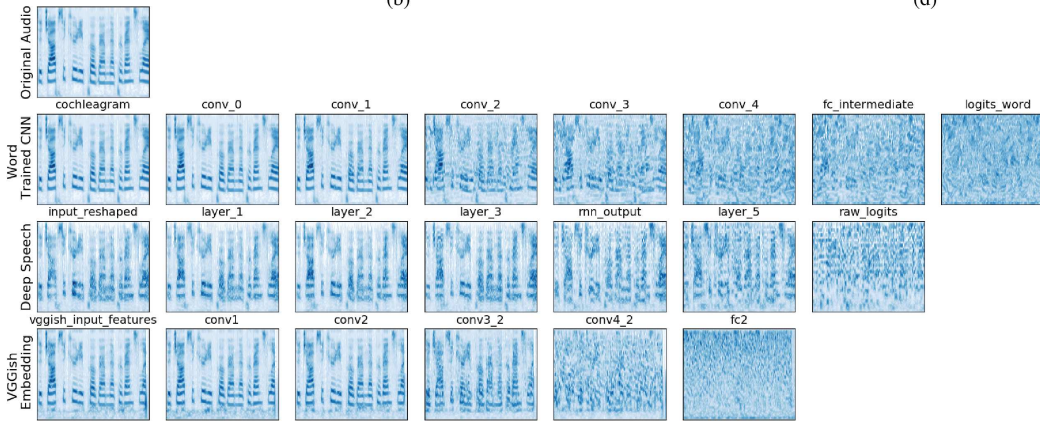

(c)

Figure 3: Deep network model metamers and their recognition by human observers. (a) Example visual network model metamers synthesized to produce the same activations at a particular layer of a particular network as the image in the top left. (b) Human recognition of visual network model metamers. Recognition is good for early-layer metamers but poor for deep-layer metamers, implying a divergence from human perceptual representations. Error bars are standard error of the mean (SEM). (c) Example cochleagrams (time-frequency decompositions) for metamers from an audio network trained to recognize words. (d) Human recognition of word-trained CNN model metamers. As for vision-trained models, recognition is good for early-layer metamers but poor for deep-layer metamers. Error bars are SEM.

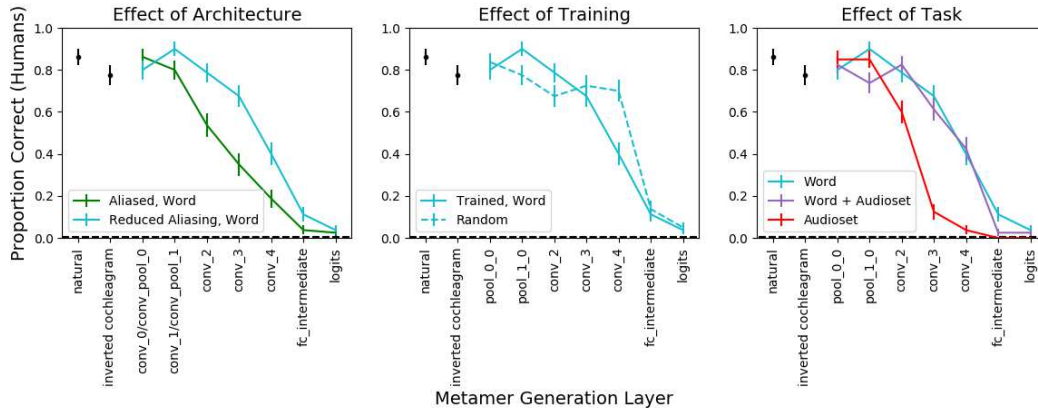

Figure 4: Human recognition of audio network model metamers. Architectural manipulations that reduce aliasing (left), training (middle), and task (right) all altered the recognizability of metamers.

a stride greater than one) there is visible 'gridding' in the metamers generated from early layers, plausibly due to aliasing.

## 4.2   Audio network model metamers

The metamers from the word-trained network with an architecture based on [8] also quickly become unrecognizable to humans (Figure 3c-d). Although not included in the human behavioral experiment, we also generated example metamers from DeepSpeech and the VGGish Embedding Network[3]. All metamers from DeepSpeech sound unnatural due to the input representation (framed MFCCs). The metamers on the VGGish embedding network become difficult to recognize by conv_4 (perhaps unsurprisingly, as we only generated metamers for speech, and the network was not trained for speech recognition).

## 4.3   Model metamers from audio networks with modified task or architecture

We considered that the decrease in metamerism for humans might be due to aliasing (from convolutional layers with strides greater than 1, and maxpooling layers). Consistent with this idea, the modified architecture that reduces aliasing yielded model metamers that were more recognizable to humans (Figure 4). We also considered the effect of training on metamerism. Unlike, the trained networks, metamers from a random network with reduced aliasing remained recognizable through all convolutional layers, only becoming unrecognizable at the top fully-connected layer. This result suggests that task optimization adds invariances to the network that can in some cases be different than human invariances. However, the human-recognizability of the model metamers was task-specific – the same network architecture trained to classify the AudioSet backgrounds produced metamers that became unrecognizable more quickly than when trained on the word task. Training on the AudioSet classification in addition to the word task did not impair metamerism (Figure 4). In all cases the metamers from deep layers remained unrecognizable to humans, but the effects of these manipulations raise the possibility that appropriate choices of training and architecture might produce a model that better accounts for human perception.

## 4.4   Metamer comparisons between ImageNet architectures

The metamer test can also be used to compare different architectures. We generated metameric images for one ImageNet-trained network and then presented its metamers to a second network. If the representational spaces between the two networks are the same, then the second network should be able to correctly classify the metamers from the first network. For all three tested networks, we find that the representations diverge from those of the other networks (Figure 5). Further, at late layers the model metamers are generally not even recognizable to the same

ImageNet-trained architecture trained with a different initialization (especially evident in the 1000 way classification task). Interestingly, image metamers for one network become non-metameric for another network at roughly the same layer at which human performance diverges.

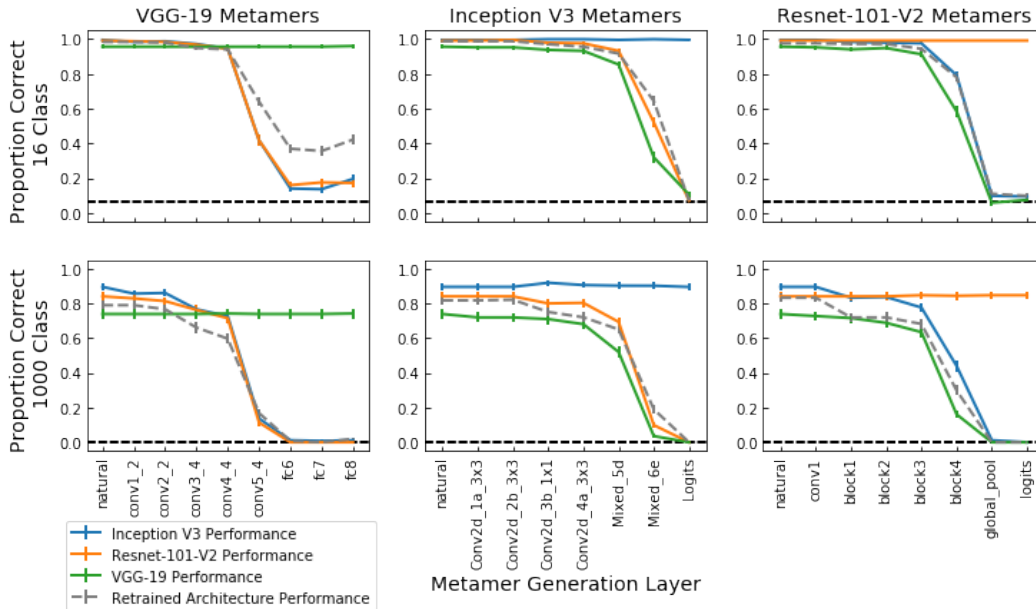

Figure 5: Network recognition of metamers from other networks and for networks with the same architecture but different initializations. All networks were trained on ImageNet. Top row: performance on 16-way classification task with metamers (using groups of the original ImageNet classes, used for human recognition experiment in Figure 2). Bottom row: performance on original ImageNet classification task with metamers.

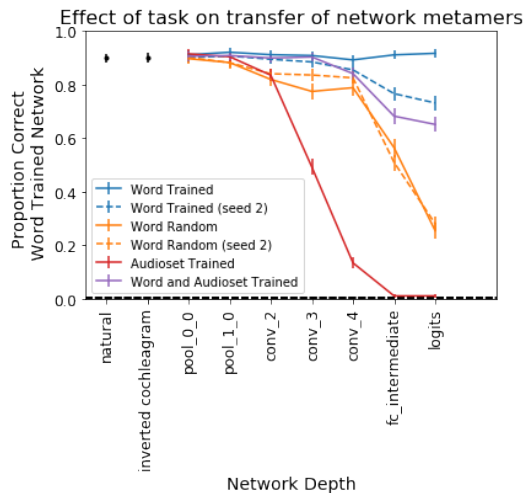

Figure 6: Word-trained network recognition of metamers from other networks. Metamers were generated from networks with the same architecture but trained on different tasks. Error bars are bootstrapped SEM.

## 4.5   Metamer comparisons between audio networks trained on different tasks

In the audio domain, we tested whether model metamers generalized across training tasks and random seeds (Figure 6). We measured performance of the word-trained network on metamers

generated from networks with the same architecture but trained on a different task. Metamers generated from untrained networks were poorly recognized by the word-trained network, providing further evidence that training alters the network invariances. Model metamerism did not transfer between the word-trained network and the AudioSet-trained network, but metamers generated from the network trained on both tasks were only slightly less metameric than metamers from a word-trained network with weights initialized with a different random seed. This latter result provides a proof of concept that it is possible for metamers to be shared across distinct systems.

## 5  Discussion

Our results show that model metamers generated from deep layers of artificial neural networks are not metameric for humans or other networks. These findings demonstrate a divergence in the invariances learned by neural networks from those present in human perceptual systems. They also highlight the benefits of using model metamers as a network comparison tool. Our results suggest that discrepancies between model and human representations, and between different models, arise in later model stages, identifying those stages as targets for model refinement. Indeed, we were able to modify some aspects of our audio-trained models to reduce aliasing and increase human recognition of the model metamers. We also demonstrate that human recognition of the metamers is dependent on the training task, possibly suggesting that the failure of humans to recognize the model metamers may be a reflection of training on a single task (in this case, recognizing speech but ignoring the background). Future work could investigate this by modifying tasks to be more diverse, or more human-like, and assessing whether the improved models better predict human behavior.

The transfer of metamers with different random seeds was surprisingly different between the image- and audio-trained networks. Further investigation revealed that optimizing the cochleagram representation rather than the audio yielded model metamers that were less recognizable by a network trained on a different random seed (Figure S9). This result raises the possibility that the shared "cochlear" pre-processing (consisting of fixed stages of convolution, pooling, and non-linearities) enforces shared invariances between audio-trained networks with different initializations. Future work could use metamerism to explore the use of shared early-layer representations as a way to unify representations across models and potentially better model human perceptual systems, for instance by adding additional biological constraints on the input representation.

Model metamers are complementary to adversarial examples. Adversarial examples are metameric (perceived similarly) for humans but are not metameric to the network they are derived for, demonstrating that the network lacks some invariances present in humans. Model metamers conversely demonstrate that invariances present in networks are not necessarily invariances for human perception (or other networks). The relationship between adversarial and metameric images was explored recently in [46], who concluded that the cross-entropy loss creates excessive invariance in the final classification layer, leading to adversarial examples. We explore related issues but examined a more diverse set of network layers and explicitly performed human and network-network experiments. Together, these lines of work suggest that techniques for reducing adversarial vulnerability may also improve the transfer of metamers across models. Moreover, metamers could be useful for evaluating the adversarial vulnerability of a model. However, unlike adversarial examples, which are specifically engineered to fool a particular system, model metamers are constrained only to produce the same model activations (rather than to fool humans). The considerable lack of metamer transfer to humans thus arguably represents a more substantial model failure, and a useful measuring stick for models of perceptual systems.

**Acknowledgements**

We thank Richard McWalter, Alex Kell, and Sam Norman-Haignere for comments on an early draft of this work. We also thank Mark Saddler and Andrew Francl for contributing to a shared codebase used in this project. This work was funded by a McDonnell Scholar Award to J.H.M., NSF grant BCS-1634050, NIH grant R01-DC017970 and a DOE CSGF Fellowship to J.J.F.

## Footnotes

[1]Example generation code and trained models: https://github.com/jenellefeather/model_metamers

[2]https://github.com/tensorflow/models/tree/master/research/slim

[3]Example audio metamers: http://mcdermottlab.mit.edu/jfeather/model_metamers/audio_metamers.html

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
