[Supplementary Material · metamers_neurips2019_FINAL_SUPPLEMENT.pdf]

## Supplement: Metamers of neural networks reveal divergence from human perceptual systems

### S1.1 Audio CNN training dataset

The auditory models were trained on the word recognition task described in [8], but with an updated training set using segments from the Wall Street Journal [39] and Spoken Wikipedia Corpora [40]. We screened the Wall Street Journal (WSJ) [39], TIMIT [46], and a subset of articles from the Spoken Wikipedia Copora (SWC) [40] for appropriate audio segments (i.e., in which words overlapped the center of a two second segment). Each segment was assigned the word class label of the word occurring at the segment midpoint, and a speaker class label determined by the speaker.

In hopes of constructing a dataset with speaker and word class labels that were approximately independent, we selected words and speaker classes such that the exemplars from each class spanned at least 50 unique cross-class labels (i.e., 50 unique speakers for each of the word classes). This exclusion fully removed TIMIT from the training dataset. We then selected words and speaker classes that each contained at least 200 unique utterances, and such that each class could contain a maximum of 25% of a single cross-class label (i.e., for a given word class, a maximum of 25% of utterances could come from the same speaker). These exemplars were subsampled so that the maximum number in any word or speaker class was less than 2000. The resulting training dataset contained 230356 unique segments in 432 speaker classes and 793 word classes, with 40650 unique segments in the validation set.

During training, the speech segments were randomly shifted in time and superimposed on AudioSet [41] examples such that models could also be trained on the AudioSet task. We randomly varied the SNR between the source (Speech) and the noise (AudioSet), uniformly distributed between -10dB SNR and 10dB SNR. To minimize ambiguity, we removed any sounds under the "Speech" or "Whispering" branch of the ontology. Since a high proportion of AudioSet clips contain music, we achieved a more balanced set by excluded any clips that were only labeled as the root "Music" with no specific branch labels, and the "Music" label was not used during the AudioSet task. We also removed silent clips by first discarding everything tagged with a "Silence" label then culling clips containing more than 10% zeros. This screening resulted in a training set of 718625 unique background clips spanning 516 categories. During training, we cycled through the sets of speech and AudioSet clips in random order, randomly sampling a two-second segment from the AudioSet clip and adding it to the speech clip to form a training example. Validation performance is reported on data constructed with the same training augmentations (specifically, variable SNR and temporal shifts). CNN models were trained across two NVIDIA GPUs each with 11GB memory.

### S1.2 Retrained ImageNet Description

The ImageNet-trained architectures used to generate metamers for the behavioral and network-network experiments were downloaded from the TFSlim repository. The code at this repository was also used to retrain ImageNet architectures for the random seed experiments. Architecture details and preprocessing were matched to the downloaded checkpoints. The batch size, number of GPUs, and learning rate that we used was likely different from that used for training the downloaded checkpoints, which is potentially reflected the slightly worse training accuracy for some of the retrained models S1.2.

| ImageNet Network | Top-1 Accuracy | Top-5 Accuracy |
|---|---|---|
| VGG-19 | 72.0 | 90.6 |
| Inception-V3 | 75.2 | 92.5 |
| Resnet-101-V2 | 73.6 | 91.5 |

Table S1: Summary of retrained ImageNet architectures for random seed experiments.

Table S2: Auditory CNN Architecture Definition ([8] with reshaped kernels to account for the modified input size.

| Layer | Type | Filters | Size | Stride |
|---|---|---|---|---|
| 0 | input | - | [211, 400] | - |
| 1 | batch-normalization | - | - | - |
| 2 | conv2d | 96 | [7, 14] | [3, 3] |
| 3 | relu (conv_0) | - | - | - |
| 4 | max-pooling2d | - | [2, 5] | [2, 2] |
| 5 | batch-normalization | - | - | - |
| 6 | conv2d | 256 | [4, 8] | [2, 2] |
| 7 | relu (conv_1) | - | - | - |
| 8 | max-pooling2d | - | [2, 5] | [2, 2] |
| 9 | batch-normalization | - | - | - |
| 10 | conv2d | 512 | [2, 5] | [1, 1] |
| 11 | relu (conv_2) | - | - | - |
| 12 | conv2d | 1024 | [2, 5] | [1, 1] |
| 13 | relu (conv_3) | - | - | - |
| 14 | conv2d | 512 | [2, 5] | [1, 1] |
| 15 | relu (conv_4) | - | - | - |
| 16 | avg-pool | - | [2, 5] | [2, 2] |
| 17 | flatten | - | - | - |
| 18 | fully-connected | 4096 | - | - |
| 19 | relu (fc_intermediate) | - | - | - |
| 20 | dropout, 0.5 | - | - | - |
| 21 | fully-connected classification (logits) | - | - | - |

Table S3: Auditory CNN Architecture Definition with Reduced Aliasing

| Layer | Type | Filters | Size | Stride |
|---|---|---|---|---|
| 0 | input | - | [211, 400] | - |
| 1 | batch-normalization | - | - | - |
| 2 | conv2d | 96 | [7, 14] | [1, 1] |
| 3 | relu | - | - | - |
| 4 | hpool (pool_0_0) | - | [12, 12] | [3, 3] |
| 5 | hpool | - | [8, 8] | [2, 2] |
| 6 | batch-normalization | - | - | - |
| 7 | conv2d | 256 | [4, 8] | [1, 1] |
| 8 | relu | - | - | - |
| 9 | hpool (pool_1_0) | - | [8, 8] | [2, 2] |
| 10 | hpool | - | [8, 8] | [2, 2] |
| 11 | batch-normalization | - | - | - |
| 12 | conv2d | 512 | [2, 5] | [1, 1] |
| 13 | relu (conv_2) | - | - | - |
| 14 | conv2d | 1024 | [2, 5] | [1, 1] |
| 15 | relu (conv_3) | - | - | - |
| 16 | conv2d | 512 | [2, 5] | [1, 1] |
| 17 | relu (conv_4) | - | - | - |
| 18 | avg-pool | - | [2, 5] | [2, 2] |
| 19 | flatten | - | - | - |
| 20 | fully-connected | 4096 | - | - |
| 21 | relu (fc_intermediate) | - | - | - |
| 22 | dropout, 0.5 training | - | - | - |
| 23 | fully-connected classification (logits) | - | - | - |

| Network Metamer Generation Layer | Number Generated Metamers | Number features | Median Spearman $\rho$ at Layer | Median Spearman $\rho$ Null at Layer | Median Spearman $\rho$ at Logits |
|---|---|---|---|---|---|
| Natural Sound | 295 | 84400 | - | - | - |
| Inverted Cochleagram | 295 | 84400 | 0.998674 | 0.179334 | 0.999976 |
| Word Trained (Aliased) | | | | | |
| conv_0 | 292 | 913344 | 0.994434 | 0.265085 | 0.999764 |
| conv_1 | 294 | 156672 | 0.980023 | 0.268980 | 0.998522 |
| conv_2 | 293 | 78336 | 0.918038 | 0.170268 | 0.997408 |
| conv_3 | 294 | 156672 | 0.956672 | 0.233818 | 0.999491 |
| conv_4 | 295 | 78336 | 0.996275 | 0.039919 | 0.999997 |
| fc_intermediate | 291 | 4096 | 0.944563 | 0.079779 | 0.999487 |
| logits | 290 | 794 | 0.995809 | 0.139888 | 0.995809 |
| Word Trained (Reduced Aliasing) | | | | | |
| pool_0 | 291 | 913344 | 0.997652 | 0.561209 | 0.999733 |
| pool_1 | 288 | 156672 | 0.989475 | 0.602778 | 0.997916 |
| conv_2 | 292 | 78336 | 0.991182 | 0.205391 | 0.999851 |
| conv_3 | 293 | 156672 | 0.989225 | 0.280117 | 0.999919 |
| conv_4 | 295 | 78336 | 0.972968 | 0.048382 | 0.999996 |
| fc_intermediate | 290 | 4096 | 0.999361 | 0.147935 | 0.999813 |
| logits | 286 | 794 | 0.998158 | 0.147180 | 0.998158 |
| Random (Reduced Aliasing) | | | | | |
| pool_0 | 272 | 913344 | 0.997214 | 0.952567 | 0.999999 |
| pool_1 | 278 | 156672 | 0.999251 | 0.962971 | 0.999997 |
| conv_2 | 281 | 78336 | 0.999756 | 0.968697 | 0.999997 |
| conv_3 | 279 | 156672 | 0.999791 | 0.963797 | 0.999997 |
| conv_4 | 285 | 78336 | 0.999814 | 0.959306 | 0.999997 |
| fc_intermediate | 289 | 4096 | 0.999683 | 0.985956 | 0.999994 |
| logits | 293 | 794 | 0.999996 | 0.986279 | 0.999996 |
| Trained Audioset (Reduced Aliasing) | | | | | |
| pool_0 | 291 | 913344 | 0.998042 | 0.451898 | 0.999866 |
| pool_1 | 290 | 156672 | 0.994289 | 0.454849 | 0.999089 |
| conv_2 | 290 | 78336 | 0.986702 | 0.193952 | 0.999923 |
| conv_3 | 291 | 156672 | 0.964322 | 0.137700 | 0.999967 |
| conv_4 | 292 | 78336 | 0.966812 | 0.134290 | 0.999972 |
| fc_intermediate | 294 | 4096 | 0.997083 | 0.314034 | 0.999972 |
| logits | 292 | 517 | 0.999752 | 0.463126 | 0.999752 |
| Trained Word and Audioset (Reduced Aliasing) | | | | | |
| pool_0 | 285 | 913344 | 0.997472 | 0.555888 | 0.999618 |
| pool_1 | 282 | 156672 | 0.990088 | 0.560066 | 0.996624 |
| conv_2 | 286 | 78336 | 0.982321 | 0.179038 | 0.999580 |
| conv_3 | 287 | 156672 | 0.976542 | 0.212300 | 0.999808 |
| conv_4 | 288 | 78336 | 0.921548 | 0.047874 | 0.999943 |
| fc_intermediate | 292 | 4096 | 0.959105 | 0.232050 | 0.999751 |
| logits | 289 | 794 | 0.998801 | 0.146840 | 0.998801 |

Table S4: Summary of network metamer generation for audio network. The number of generated network metamers varies by layer due to failed optimizations (measured by an overlap with the null or not having the same maximum logit as the original) or due to node time outs during the generation. Null distributions are constructed from 1,000,000 image pairs in the training set. Metamers included in the experiment do not overlap with the null distributions, even in the case of the Random (Reduced Aliasing) network layers where activations are strongly correlated for the null. Metamers were generated on NVIDIA GPUs with 11-12GB of RAM.

| Network Metamer Generation Layer | Number Generated Metamers | Number features | Median Spearman $\rho$ at Layer | Median Spearman $\rho$ Null at Layer | Median Spearman $\rho$ at Logits |
|---|---|---|---|---|---|
| Natural Image | 256 | 89401 | - | - | - |
| Natural Image Small | 256 | 50176 | - | - | - |
| Inception-V3 [43] | | | | | |
|    Conv2d_1a_3x3 | 256 | 710432 | 0.999980 | 0.724671 | 1.000000 |
|    Conv2d_2b_3x3 | 256 | 691488 | 0.999539 | 0.510215 | 0.999994 |
|    Conv2d_3b_1x1 | 244 | 426320 | 0.984236 | 0.335206 | 0.990008 |
|    Conv2d_4a_3x3 | 253 | 967872 | 0.995720 | 0.592758 | 0.998891 |
|    Mixed_5d | 254 | 352800 | 0.992983 | 0.183679 | 0.999667 |
|    Mixed_6e | 253 | 221952 | 0.950064 | 0.172391 | 0.998504 |
|    Mixed_7c [3] | 240 | 180224 | 0.756891 | 0.064566 | 0.961890 |
|    Logits | 255 | 1001 | 0.999831 | 0.040540 | 0.999831 |
| Resnet-101-V2 [44] | | | | | |
|    conv_1 | 256 | 1440000 | 1.000000 | 0.120331 | 1.000000 |
|    block_1 | 256 | 369664 | 0.999787 | 0.754825 | 0.999448 |
|    block_2 | 256 | 184832 | 0.999978 | 0.496263 | 0.999981 |
|    block_3 | 254 | 102400 | 0.999302 | 0.342142 | 0.999609 |
|    block_4 | 255 | 204800 | 0.994098 | 0.284898 | 0.999230 |
|    global | 254 | 2048 | 0.902678 | 0.214380 | 0.998909 |
|    logits | 254 | 1001 | 0.999659 | 0.047858 | 0.999659 |
| VGG-19 [42] | | | | | |
|    conv1_2 | 256 | 3211264 | 0.999961 | 0.184170 | 1.000000 |
|    conv2_2 | 256 | 1605632 | 0.999152 | 0.066985 | 0.999998 |
|    conv3_4 | 256 | 802816 | 0.999155 | 0.108890 | 0.999995 |
|    conv4_4 | 255 | 401408 | 0.994657 | 0.035149 | 0.999994 |
|    conv5_4 | 256 | 100352 | 0.971722 | 0.022134 | 0.999980 |
|    fc6 | 256 | 4096 | 0.977821 | 0.031115 | 0.999993 |
|    fc7 | 256 | 4096 | 0.987343 | 0.043484 | 0.999980 |
|    fc8 (logits) | 255 | 1000 | 0.999924 | 0.187791 | 0.999924 |

Table S5: Summary of network metamer generation for visual networks. [1] Although metamers were generated for Mixed_7c, we did not include Mixed_7c metamers for human behavior or model-model comparisons, as the optimization did not succeed to the same extent as the other layers (detailed histogram in Figure S6)

Figure S1: Spearman correlation coefficient for the word task CNN metamer generation compared with a null correlation distribution obtained by correlating 1000000 random speech sounds from the training set. Diagonal elements (with figure titles in red) correspond to the network metamer generation layer. For a given metamer generation layer, metamer Spearman correlations for the later network layers (further to the right) remain far from the null, while for earlier layers the distributions begin to overlap with the null, demonstrating the the generated stimulus is physically distinct from the natural sound.

Figure S2: Network metamer Spearman correlation coefficients compared with the null correlation distribution for the Work Task CNN (with reduced aliasing).

Figure S3: Network metamer Spearman correlation coefficients compared with the null correlation distribution for the Random Word Task CNN (with reduced aliasing). Even though the null distribution correlations are very high for deep layers in this network, there is no overlap between the null distributions and the distribution from model metamers used for the experiments.

Figure S4: Network metamer Spearman correlation coefficients compared with the null correlation distribution for the Audioset Task CNN (with reduced aliasing).

Figure S5: Network metamer Spearman correlation coefficients compared with the null correlation distribution for the Word and Audioset Task CNN (with reduced aliasing).

Figure S6: Model metamer Spearman correaltion coefficients compared with the null correlation distribution for Inception-V3. Metamers were generated for layer Mixed_7c, however the optimization did not succeed to the same extent as the other layers (with a median Spearman $\rho$ below 0.9) and we thus do not report behavioral results for this layer.

Figure S7: Network metamer Spearman correlation coefficients compared with the null correlation distribution for VGG-19.

Figure S8: Network metamer Spearman correlation coefficients compared with the null correlation distribution for Resnet-V2-101.

Figure S9: Transfer of metamers between the same architecture and task but different random seeds when generating the metamer by optimizing the waveform (as in all our main experimental conditions, because we needed to present the stimuli as sounds to human listeners) vs. the cochleagram. The audio waveform-generated metamers transfer between two architectures trained on different random seeds, while the cochleagram-generated metamers do not. This suggests that including the cochleagram generation stages in the optimization imposes additional constraints on the audio that restrict the representational capacity, increasing the likelihood of transfer across models. Quality of cochleagram metamer generation is summarized in Table S1.2

| Network Metamer Generation Layer | Number Generated Metamers | Number features | Median Spearman $\rho$ at Layer | Median Spearman $\rho$ Null at Layer | Median Spearman $\rho$ at Logits |
|---|---|---|---|---|---|
| Natural Sound | | | | | |
|    orig | 295 | 84400 | - | - | - |
|    visualization | 295 | 84400 | 0.998674 | 0.179334 | 0.999976 |
| Word Trained (Reduced Aliasing), Cochleagram Metamers | | | | | |
|    pool_0 | 293 | 913344 | 0.999119 | 0.561209 | 0.999974 |
|    pool_1 | 292 | 156672 | 0.998646 | 0.602778 | 0.999982 |
|    conv_2 | 293 | 78336 | 0.998229 | 0.205391 | 0.999989 |
|    conv_3 | 294 | 156672 | 0.995979 | 0.280117 | 0.999976 |
|    conv_4 | 293 | 78336 | 0.983156 | 0.048382 | 0.999993 |
|    fc_intermediate | 290 | 4096 | 0.992512 | 0.147935 | 0.998165 |
|    logits | 281 | 794 | 0.995095 | 0.147180 | 0.995095 |

Table S6: Summary of network metamer generation for audio metamers generated by optimizing the cochleagram.