[Reviews · NeurIPS 2019]

Reviewer 1



Updated Review: Thanks to the authors for clarifying the stopping criteria for meatier synthesis. My review for this work remains the same following author response because I believe that the authors have demonstrated this work to be of high quality and relevance to the NeurIPS community. Originality: Although the algorithms used to synthesize the metamers themselves are nothing new, the work is a novel combination of previous approaches and techniques, and the analysis approach gives these methods a fresh perspective that leads to good insights. Quality: The work is of high quality and is a complete piece of work that will advance our understanding of the relationships between architecture, task and training in determining representational similarity between networks and humans (as well as between networks). Clarity: The paper is overall clear, though some details could use a bit of clarifying (what was the threshold for satisfactory termination of synthesis? How was this determined?) Significance: This work builds on theoretical and experimental work from neuroscience used to analyze how well models of perceptual systems capture the representation within the human brain by synthesizing stimuli that match the responses of some part of the model completely and using human subjects to validate that the original and matched stimulus are in fact the same. The authors adapt this test for use in comparing the representations in neural network models trained on complex tasks with human representations helping to bridge the two fields. This work also builds on a string of recent work attempting to find similarities and differences between our learned model representations and human perceptual representations, and adds a critical analysis tool for understanding network deviations. The test stated here is actually looser than the original metamer test, which required not that the matched images or sounds were classified the same (which is permissive of some perceptual deviation or distortion), but were perceptually indistinguishable. Nonetheless, I believe this is a useful tool for comparing representational similarity, both between networks and humans, and between different types of networks and allows the authors to extract meaningful empirical contributions isolating architectural and training modifications that bring the representations “more in line” with human representations (accounting for aliasing in architecture, trained vs untrained networks, and training on specific databases and tasks vs others). Followup work with this technique could help identify a wider corpus of tasks that lead to metameric similarity between networks and humans and can lead to deep insight about the goals that human perceptual systems are optimized for.

Reviewer 2



Updated after author response ---------------------------------------- Thanks to the authors for their response. In light of their points, and the other reviews, I have increased my score to a 7. Please include the random seed ImageNet results in the final revision! Original review -------------------- Overall, I enjoyed reading this paper. The authors provide a very thorough set of experiments that investigates the use of metamers for probing artificial neural networks. I appreciated that the authors took the time to carry out the human experiments, generated metamers for networks both in the visual and auditory domains, and tested multiple architectures in each setting. Major comments: While the methods are solid and the results comprehensive, I have some questions regarding the interpretation of the results. The main finding seems to be that metamers of deeper layers of artificial networks are not perceptually recognizable by humans. My reading of the paper is that the authors interpret this as a failing of artificial networks as a model of human perception. For example, they state on lines 38-39 that they "leverage the idea that metamers for a valid model of human perception should also be metamers for humans". I disagree with this assessment, for a few different reasons: - First, there may be many purposes for models of human perception, not all of these require that the model match human perception exactly (i.e. all models are wrong, but some are useful). - Second, the authors are comparing networks trained to perform object classification (a specific visual task) or word recognition (a specific auditory task) with the entirety of human visual and auditory perception, which needs to support a multitude of perceptual tasks relevant for behavior. For example, in vision, perhaps a better comparison would be to test whether the metamer is sufficient for just the ventral visual pathway (although I understand this is experimentally infeasible). Regardless, I do not see why we should expect that networks trained on a single task should generate perceptual metamers recognizable by humans; as these networks are only trained to solve a very specific subset of what the human perceptual system is required to do. - Third, I want to offer a different interpretation of the results. Imagine a hypothetical experiment where one is able to perform the same optimization (synthesizing a metamer) for a particular human, by optimizing it to match the activations in some part of the brain (pretend we can access all of the relevant neurons and synaptic weights in that brain). Would the synthesized metamer be recognizable by other humans? I am not so sure--perhaps optimizing the input for a particular brain would make it unrecognizable to other brains. The analogy in artificial networks would be to see if synthesizing a metamer for one network are recognizable by other networks. Indeed, the authors did some of these experiments--and find that they are not! So perhaps expecting metamers to be recognizable across networks (either human or artificial) is too stringent a requirement. Minor comments: - The first paragraph of the introduction discusses some of the ways in which deep networks have been shown to be different from human perception. The paragraph ends with the statement that "the consequences of these discrepancies ... remains unclear". However, it feels to me that this paper does not address this question, instead, it seems to add more discrepancies (in the form of metamers), but does not take on the question of understanding the implications of those discrepancies. (I would simply suggest rewording the first intro paragraph to focus on what the paper focuses on). - The authors comment on difficulties associated with matching synthesizing metamers for units after ReLU nonlinearities. Instead of the proposed solution, why not simply target the unit activations *before* the nonlinearity?

Reviewer 3



After author response: I might have been a bit harsh in my initial quantitative assessment and now corrected my overall score accordingly. The authors do correctly point out that earlier papers do not even report how closely they actually fitted the model representations with the generated metamers and thus might well suffer from similar problems as I described. Also, the paper does provide evaluations on both vision and audition and comparisons to humans and between different networks, which is definitely a broad set of results, which I somewhat undervalued in my initial review. I still think that most results presented in this paper could be expected based on earlier literature. Nonetheless, I would definitely no longer argue for a rejection. -------------------------- This manuscript presents evaluations for networks which recognize words from speech and for ImageNet trained object recognition networks. In both cases the authors optimize stimuli from white noise to match the representation in a specific layer of the network to generate a metamer for the network. These metamers are then shown to humans or other networks who perform the same task. Generally stimuli are well recognizable and very similar to the original stimulus for early layers and then deteriorate into noise for higher levels, which cannot be recognized by humans or other networks. The ends of this spectrum were known: low layers loose little information, i.e. metamers are very close to the original stimulus and the final layers are subject to noise adversarial examples, i.e. there are noise patches which give any logits. Nonetheless a more detailed comparison where this change happens and whether there is a range in-between is an interesting question. Also the authors present experiments on different networks and for both vision and audition. However, I have two larger criticisms of this paper: 1)The stimuli used are not really metamers for the model layers they were trained for. Although the authors introduced soft ReLUs to get closer the activations they produce are sometimes only .9 correlated with the model activations, which is not optimal, as there clearly is the original image, which got the exactly right activations. As the optimization also seems to be harder for deeper layers this introduces an interpretation issue, how much of the effect is due to imperfect optimization. 2) Started by the papers [15] & [16] cited in the paper there is a literature trying to explain crowding, i.e. metamerism in the periphery by deep neural network representations. Papers on this have done similar experiments to the ones in this paper, although they introduced pooling in the periphery and asked whether images are metameric for humans as well instead of only requiring equal categorization. Example papers include: Deza, A., Jonnalagadda, A., & Eckstein, M. (2017). Towards Metamerism via Foveated Style Transfer. ArXiv:1705.10041 [Cs]. Retrieved from http://arxiv.org/abs/1705.10041 https://openreview.net/forum?id=BJzbG20cFQ Wallis, T. S. A., Funke, C. M., Ecker, A. S., Gatys, L. A., Wichmann, F. A., & Bethge, M. (2017). A parametric texture model based on deep convolutional features closely matches texture appearance for humans. Journal of Vision, 17(12), 5. https://doi.org/10.1167/17.12.5 Wallis, T. S., Funke, C. M., Ecker, A. S., Gatys, L. A., Wichmann, F. A., & Bethge, M. (2019). Image content is more important than Bouma’s Law for scene metamers. https://doi.org/10.7554/eLife.42512 These seem highly relevant for the comparison done in this paper and I think a discussion what is similar or different is warranted. Overall, I thus believe this manuscript does not contribute enough new insights, given the technical interpretation problem and that there was considerable knowledge about the curves measured beforehand.

[Author Response · NeurIPS 2019]

We thank the reviewers for their thoughtful comments and suggestions. We are happy that reviewers see this line of work as useful for both neuroscientists and ML researchers. Here we paraphrase and respond to the main criticisms.

**The results could reflect failures of optimization(R1, R4)** R1 had questions about the criterion for deciding that a stimulus was a model metamer. R4 supposed that the optimization is harder for deeper layers and wondered whether this could explain the variation in human-recognizability with depth.

As described in section 3.1, once we ran the optimization procedure for 15,000 iterations, the following two conditions had to be true for a model metamer to be included in our experiments: (1) The network predicted the same label for the synthetic metamer and the paired natural image. This is the same classification test we apply to humans and other networks. (2) The spearman $\rho$ fell outside of the null distribution of activations measured between two randomly chosen image pairs (Supplement Figs 1-8). We will clarify the motivation for this approach in the revision. We believe that the comparison to a null distribution from random inputs is better than applying a strict threshold, because the expected correlation varies with network layer. Setting hard cutoffs could potentially call samples metameric which are no more matched than chance. Empirically, we found this approach crucial for random networks (Supplement Fig 3).

Regarding the potential confound described by R4: the quality of optimization is in general not worse in deeper layers. Indeed, the final layer of logits is among the easiest to match via optimization (all with median spearman $\rho$ above 0.99, Supplement Tables 3 and 4) and all tested models have model metamers generated from the logits that are unrecognizable. Further, for metamers generated at other layers of the network, the corresponding logits are highly correlated with those of the original input (median spearman $\rho > 0.99$), i.e., the model is nearly equally confident in its prediction for the natural and the synthetic stimulus. We will include these median logit correlations as another column in Supplemental Tables 3-4 in the final manuscript. We agree that additional optimization improvements are of interest for future work (the linear gradient ReLUs used here contribute to this effort). However, because we required the model to predict the original class label for each metamer, the substantial lack of recognition by humans point to significant discrepancies between the model and human representations even if the metamer representations are not "exactly" the same. We will rephase lines 39-43 to capture this important point. We also note that similar past work [18, 41] did not quantify the degree to which the optimization succeeded, and we believe our metrics are a step forward.

**We need a more nuanced take on model success and failure (R2)** We largely agree. We will rework the introduction and discussion to emphasize that what we take as model failures may not apply to all lines of work. But it seems likely that model metamers could guide discovery of neural networks that more closely resemble human perception.

**Failure of the metamer test could reflect training on a single task (R2)** We agree models will no doubt be limited by training on a single task. However, we contend that a model intended to replicate the basis of human performance of a particular task, such as speech recognition, should have metamers that are at a minimum recognizable to humans, even if they do not sound/look exactly the same as the original (potentially due to the single-task training of the model). We examined the effect of training task in Figure 4, and argue that more human-relevant tasks can improve models, but we will clarify this issue in the final paper.

**Transfer of metamers across random initializations (R2)** For the audio-trained networks, metamers generally transferred across initializations. ImageNet random seed results were not included in our manuscript because we used publicly available checkpoints that only had one training run. We have since begun training ImageNet architectures with two different random seeds to ask precisely this question, and will include those data in Figure 5 of the final paper.

**The relation to Jacobsen et al. [41] is unclear (R2)** [41] focuses on how "excessive invariance" (model metamers that are not classified the same by humans) relate to adversarial examples, and propose a modification to the cross entropy loss to reduce invariance. The general conclusions are similar to ours, but they exclusively study the final classification layer, and in qualitative terms. We instead examine all layers and explicitly perform human and network-network experiments, investigating how task and architecture can shape the space of network invariance. Further, [41] is focused on reducing adversarial vulnerability, while our motivation is to introduce metamers as a generic model comparison tool. We will include another sentence in the discussion explicitly addressing the relationship to [41].

**The relation to visual crowding literature is unclear (R4)** We will add a related work section further elaborating on this literature in the final paper. As R4 notes, this literature uses pooling in the periphery of visual models to test how well their features align with a particular aspect of human perception. We see our work as a more general instantiation of this approach, applicable to domains outside of peripheral vision where invariances arise in the service of recognition, rather than as a consequence of pooling. Moreover, our work introduces metamers for artificial model comparison, which is highly relevant to the ML community.

**Work on metamers that are especially different from the original (R4)** We agree that this approach could be useful. However, our paper shows that even without adversarial constraints, model metamers are often unrecognizable to humans. We believe this is a more generic model failure than those obtained from stimuli that are explicitly distinct.

[Meta-Review · NeurIPS 2019]

Dear authors, congrats on the acceptance-- this paper was discussed extensively, the the reviewers provided multiple comments and feedback-- please do take the feedback in the reviews into account when preparing your final manuscript.